# Factors influencing TB treatment interruption and treatment outcomes among patients in Kiambu County, 2016-2019

Evelyn Kimani[1]*, Samuel Muhula[2], Titus Kiptai[2], James Orwa[3], Theresa Odero[3], Onesmus Gachuno[3]

1 Department of Health, Tuberculosis, Leprosy and Lung Disease Program-Kiambu County, Kiambu, Kenya, 2 AMREF Health Africa in Kenya, Nairobi, Kenya, 3 University of Nairobi – School of Medicine, Nairobi, Kenya

⊙ These authors contributed equally to this work.
* drevelyne@gmail.com

**Data Availability Statement:** All relevant data are within the manuscript and its Supporting information files.

## Abstract

Tuberculosis (TB) is the leading cause of mortality as a single infectious agent globally with increasing numbers of case notification in developing countries. This study seeks to investigate the clinical and socio-demographic factors of time to TB treatment interruption among Tuberculosis patients in Kiambu County, 2016–2019. We retrospectively analyzed data for all treatment outcomes patients obtained from TB tracing form linked with the Tuberculosis Information Basic Unit (TIBU) of patients in Kiambu County health facilities using time to treatment interruption as the main outcome. Categorical variables were presented using frequency and percentages. Kaplan-Meir curve was used to analyze probabilities of time to treatment interruptions between intensive and continuation phases. Log-rank test statistics was used to compare the equality of the curves. Cox proportion model was used to determine determinants of treatment interruption. A total of 292 participants were included in this study. Males were 68%, with majority (35%) of the participants were aged 24–35 years; 5.8% were aged 0–14 years and 5.1% aged above 55 years. The overall treatment success rate was 66.8% (cured, 34.6%; completed 32.2%), 60.3% were on intensive phase of treatment. Lack of knowledge and relocation were the major reasons of treatment interruptions. Patients on intensive phase were 1.58 times likely to interrupt treatment compared to those on continuation phase (aHR: 1.581; 95%CI: 1.232–2.031). There is need to develop TB interventions that target men and middle aged population in order to reduce treatment interruption and increase the treatment success rates in the County and Country.

## Introduction

Tuberculosis (TB) remains a major public health concern globally. Though an ancient disease, it is still among the top ten causes of mortality globally from a single infectious agent [1]. In 2018, the estimated number of TB cases were 10 million [1] worldwide with 1.2 million estimated among the Human Immunodeficiency Virus (HIV) negative patients and 251,000

**Funding:** The author(s) received no specific funding for this work. The funders had no role in study design, data collection and analysis, decision to publish, or preparation of the manuscript.

**Competing interests:** The authors have declared that no competing interests exist.

among TB-HIV co-infected patients [1]. The number of TB cases notified worldwide in 2018 were notably higher among the male gender at 56% [1].

Kenya, a low middle income country is among 30 high burden countries for TB and among 14 countries that suffer a triple burden of TB, TB-HIV and Multi-drug resistant (MDR) TB clients [1–3]. In 2018 alone, the country reported 96,478 TB cases; a 13.2% increment from the previous year. In the same year, TB-HIV co-infection notified cases contributed 26.6% [4]. The country has come a long way in enhancing TB diagnosis through equitable distribution of diagnostic tools like expanded use of x-ray, availability of gene-xpert machines as well as provision of microscopy for both diagnosis and follow-up for TB patients. Further, initiatives are still in place in bridging the gap of 40% missing TB cases reported by the TB prevalence survey of 2015-2016 [5]. Some of the initiatives include health facility active case finding, contact tracing by Community Health Volunteers (CHVs), public private partnerships as well as Kenya Innovation Challenge for TB. All these are aimed at finding missing people with TB in the communities and linking them to TB diagnosis and treatment clinics through innovative strategies.

The strategic development goal (SDG) number three aims to improve good health and well-being among all people with one of the targets being to end TB and HIV epidemics globally by 2030 [1]. Similarly, the End TB strategy aims at reducing the number of deaths due to TB by 90% as well as reduce the TB incidence by 80% by 2030 [4]. Likewise, the Kenya National TB programme aims at ensuringthat all the TB notified patients adhere to the recommended regimen and successfully complete treatment. These milestones will not be realized if the challenges facing treatment interrupters are not adequately addressed.

In a bid to strengthen retention to care among the TB patients, Kiambu County, with the support from Respiratory Society of Kenya (RESOK) through Amref Health Africa, has engaged community health volunteers (CHVs) across the health facilities offering diagnostic and/or treatment services to TB patients. With support from the health care workers (HCWs), CHVs actively follow up patients who either have missed an appointment or interrupted treatment for more than 2 weeks. The treatment interrupters are documented in the Ministry of Health (MOH) community reporting tool- treatment interruption tracing form.

To further improve the adherence, the National TB programme in Kenya adopted the Direct Observation Therapy (DOT) for TB patients as recommended by the World Health Organization (WHO) in 1993 [6] where all patients with TB disease are supervised while taking their medication. However, there are patients that fail to keep their appointment dates hence missing their treatment. The interruption to treatment could range from weeks to months. During this time, the patients risk transmitting TB disease to their close contacts, could develop resistance due to drug interruption or end up with unfavorable outcomes such as death.

Kiambu county in 2018, had a notable 5.9% of patients interrupting their treatment either during intensive phase or continuation phase, a percentage that is above the national level of 5.4% [7]. A study conducted in Meru Kenya, found that more males interrupted treatment and cited forgetfulness as the commonest reason for treatment interruption [8]. In addition, a study conducted in Zambia revealed the commonest reasons for treatment interruption being feeling better, low level of knowledge on treatment completion benefits and drug side effects. Other contributory factors found among treatment interrupters in Uganda was change of the treatment facility for the patient [9]. Social factors found were lack of formal employment, lack of family support as well as smear positivity at time of diagnosis [10]. Other factors identified in a study conducted in South Sudan were long distance to the health facility, stigma from the society, high cost of transportation, traditional beliefs as well as rural residency [11]. In North-West Ethiopia, a cross-sectional study conducted revealed that patients with more than one

co-morbidity, poor patient-provider relationship and low knowledge on TB contributed significantly to treatment interruption [12]. A systematic study review on qualitative research conducted on patient adherence to tuberculosis treatment identified areas that included financial burden, law and immigration, knowledge, attitude, beliefs, family, community and household support [13] as being key factors to treatment interruption.

There is paucity of data on treatment interruption and outcomes in Kiambu County. The study will bridge the information gap by investigating the socio-demographic and determinants of time to interruption and treatment outcomes between 2016 and 2019. This information will be vital in strengthening the health system in the County and every effort put to ensure the barriers highlighted are addressed to improve the treatment success among the TB patients even as the government strives to reduce the number of new TB infections by 90% by 2030.

## Methods

### Data collection

The data for this study were extracted from the MOH Tuberculosis treatment interruption tracing form attached as additional file (S1 File) for the period 2016 to 2019 which captured the following variables: age, sex, location of health facility treatment site and type of TB that the participant is suffering from, date of treatment initiation, date of treatment interruption and tracing outcomes.

The data in the MOH TB tracing form is collected by trained CHVs as they follow up patients who interrupt treatment. The patients who are traced by the CHVs are referred back to the health facility and linked to the Community Health Extension Worker (CHEW) for continuation of care and treatment. The information captured in the tracing form is submitted to Respiratory Society of Kenya (RESOK) for entry into the Grants Management Information System (GMIS) by the data clerks and data manager based at Amref Health Africa. The data entered is verified against the physical form to ensure accuracy in entry. The treatment outcome data was obtained by linking the data from GMIS and Tuberculosis Information Basic Unit (TIBU) system.

### Study design

This was a retrospective study of TB patients in Kiambu County health facilities who interrupted treatment, traced and successfully referred back to treatment. The study period was from 1stJanuary 2016 to 31stDecember 2019.

### Study site and participants

Kiambu County, one of the 47 Counties in Kenya, has a population of 2.4 million people and is ranked among the top ten TB high burden counties in the country. It is divided into 12 administrative sub-counties with a total of 505 health facilities. Of these, 108 (21%) are public health facilities, 64 (13%) are faith based organizations and 333 (66%) are private facilities. Among the public health facilities, there are 70 level two, 24 level three, 11 level 4 and three level 5 health facilities [14]. Level 2 and 3 health facilities offer basic medical services while level 4 and 5 facilities offer tertiary health services. The study was conducted in level 3, 4 and 5 health facilities in Kiambu County. Of all the health facilities in Kiambu County, there are 142 treatment sites for TB, 82 diagnostic sites and 7 gene-Xpert machines for diagnosis purposes.

The study population included all TB patients who interrupted treatment but were returned to care and treatment both from the public and private health sector across the12 sub-counties

of Kiambu County. There was no formal sample size calculation as this was a retrospective study, however the final sample included in the study was assumed to have enough power to detect a difference between the outcome and explanatory variables. We excluded all TB patients who did not interrupt treatment, already active in care and those who could not be traced back into the treatment.

## Sampling procedure

All the patients interrupting treatment in the period of 2016to 2019 and were successfully traced formed the study sample.

**Study variables.** The dependent variable for this study was time to treatment interruption measured as time from TB treatment initiation to the time the participant interrupted treatment. The independent variables included sex of the respondents (male and female), age at the time of data collection measured on a continuous scale, BMI categorized into obese, normal, moderate, overweight and severe; HIV status measured on binary scale as positive and negative; type of TB (PTB and EPTB) and TB treatment phase (continuation and intensive). All variables were considered important in explaining the outcome of interest and were therefore adjusted for in the Cox proportion model.

## Statistical analysis

All the data extracted from GMIS and TIBU were exported into an excel database. Categorical data were expressed as frequency and percentage. Kaplan-Meir curve was used to analyze the probability of time to treatment interruption for patients on intensive and continuation phase of treatment. The equality of the curves was compared using Log-Rank, Breslow or Tarone-Ware test statistics. Cox proportion hazards model was performed to determine socio-demographic and clinical factors associated with the time to treatment interruption both for Univariate and multivariate models. All statistical Analysis was conducted using STATA version 15 (Stata Corporation, College Station, Texas, USA) and all tests were two-tailed with p-values $< 0.05$ considered statistically significant.

## Ethical considerations

The study was approved by the Amref health Africa in Kenya scientific and ethical review committee by providing a waiver of consent because this was a retrospective study of routinely collected data; application approval P796/2020. Strict confidentiality was ensured throughout the study period and final analysis dataset de-identified prior to analysis. The study was conducted in accordance with the Declaration of Helsinki.

## Results

### General characteristics

We included 292 participants who interrupted TB treatment in Kiambu County health facilities in the period of 1st January 2016- 31st December 2019. Males were 200(68%) while female were 92 (32%). Majority (35%) of the study participants were aged 25–34 years, followed by those aged between 35–44 years (26%). The age groups 0–14 years and above 55 year constituted 5.8% and 5.1% respectively. Most of the participants were from Kiambaa (23%), Githunguri (21%) and Thika (20%) sub-counties. Among the study participants, Pulmonary TB was the most common type of TB (98.6%, n = 288) and the rest (n = 4) had extra pulmonary TB. Of the participants with Pulmonary TB, 265(92.0%) showed positive smear results while the rest (n = 3) showed smear negative. Majority (43%) of the patients had normal BMI, followed

by those with moderate BMI (30.8%), 16.8% had severe BMI, and the rest (9%) were either obese or overweight. The proportion of TB-HIV co-infection was estimated to be 29% (n = 85) while 60.3% (n = 176) of the participants were in intensive phase of the treatment as shown in Tables 1 and 2.

## Outcomes of treatment

Fig 1 shows the treatment outcomes of the patients. The overall treatment success rate (TSR) was 66.8 (either cured (34.6%) or completed (32.2%), 20.9% were lost to follow-up, 5.8% transfer out, 4.1% died, not completed treatment, non-TB was each 1%, and 0.3% were treatment failures.

## Reasons for treatment interruption

The most common reason for the treatment interruptions were lack of knowledge (17.8%) followed by relocation (17.1%), travelled (12.0%) felt better(11.0%), and other reasons are summarized in Fig 2. Others inlcudes: financial contraints, drug resistant cases, scared, long distance each at 0.3%, worsening condition, and pill burden each at 0.7%.

## Kaplan Meir survival curves for treatment interruptions

Fig 3 shows the Kaplan-Meir curves for the cumulative risk of treatment interruptions between intensive and continuation phase of the treatment. The interruption probability at the

**Table 1. Socio-demographic characteristics of TB patients attending public hospitals in Kiambu County, 2015–2019 (N = 292).**

| Variable | Frequency | Percent |
|---|---|---|
| Sex | | |
| Male | 200 | 68.5 |
| Female | 92 | 31.5 |
| Age-group in years | | |
| 0–4 | 8 | 2.7 |
| 5–9 | 4 | 1.4 |
| 10–14 | 5 | 1.7 |
| 15–24 | 49 | 16.8 |
| 25–34 | 101 | 34.6 |
| 35–44 | 75 | 25.7 |
| 45–54 | 35 | 12 |
| 55–64 | 8 | 2.7 |
| 65+ | 7 | 2.4 |
| Sub-county of residence | | |
| Gatundu | 25 | 8.6 |
| Githunguri | 60 | 20.6 |
| Juja | 18 | 6.2 |
| Kabete | 15 | 5.1 |
| Kiambaa | 66 | 22.6 |
| Kiambu | 5 | 1.7 |
| Kikuyu | 9 | 3.1 |
| Lari | 15 | 5.1 |
| Limuru | 12 | 4.1 |
| Ruiru | 10 | 3.4 |
| Thika | 57 | 19.5 |

**Table 2. Clinical characteristics of TB patients attending public hospitals in Kiambu County, 2015–2019 (N = 292).**

| Variable | Frequency | Percent |
|---|---|---|
| Type of TB | | |
| PTB | 288 | 98.6 |
| EPTB | 4 | 1.4 |
| Initial smear(n = 288) | | |
| Smear+ | 265 | 92.0 |
| Smear- | 23 | 8.0 |
| BMI | | |
| Normal | 126 | 43.2 |
| Moderate | 90 | 30.8 |
| Obese | 17 | 5.8 |
| Overweight | 10 | 3.4 |
| Severe | 49 | 16.8 |
| HIV status | | |
| Negative | 207 | 70.9 |
| Positive | 85 | 29.1 |
| Phase | | |
| Intensive | 176 | 60.3 |
| Continuation | 116 | 39.7 |

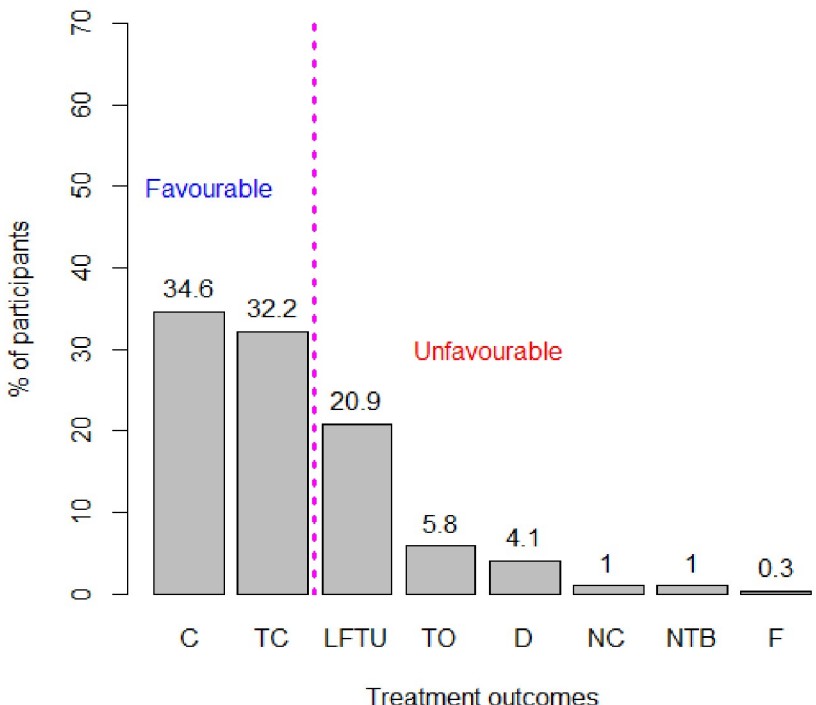

**Fig 1. TB treatment outcomes.**

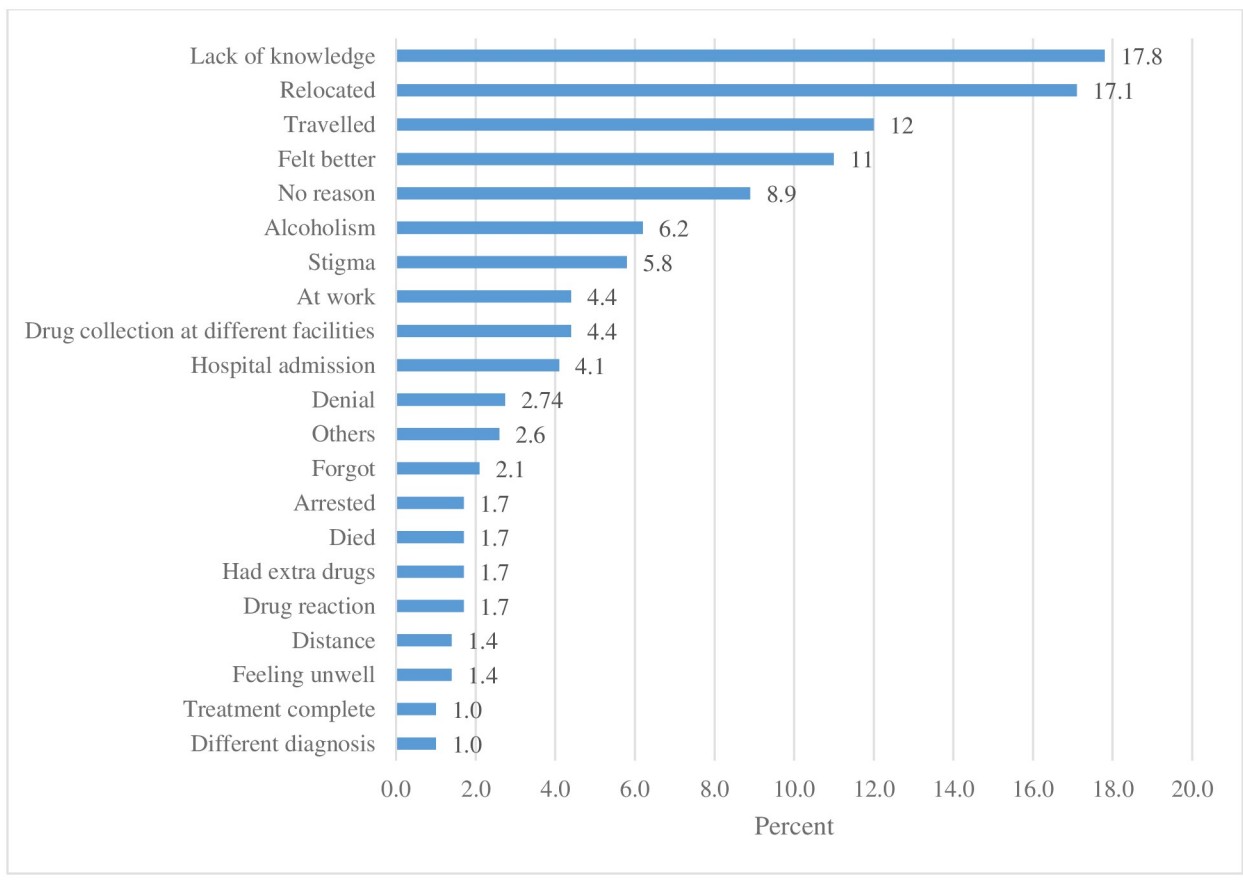

**Fig 2. Reasons for missed visits.**

intensive phase was higher than of those in the continuation phase. As time increases the two probabilities were the same. The log rank test showed that there was a significant difference between the two curves as shown by the different tests.

The results of Cox regression survival model are summarized in Table 3; only phase of treatment was significantly associated with the time to treatment interruption. Patients on intensive phase of treatment were 1.58 times more likely to interrupt treatment compared to those on continuation phase (aHR: 1.581; 95%CI: 1.232–2.031). Sex, type of TB, BMI, and HIV status were not statistically significant.

## Discussion

Treatment interruption in Kiambu County was shown to be prevalent during the intensive phase compared to continuation phase. This could be due to insufficient health education given to the participants upon being diagnosed with TB. These results are consistent with studies done in health facilities in south Ethiopia and Nairobi County that revealed that most of the patients interrupted treatment during the intensive phase of treatment [15,16]. Further, majority of the participants who interrupted treatment had pulmonary TB with a smaller proportion having extra-pulmonary TB. This could be due to the transmission mode of TB being airborne resulting to the lung parenchyma being most affected by the disease [17]. In addition, majority of the patients were bacteriologically confirmed cases at the time of treatment

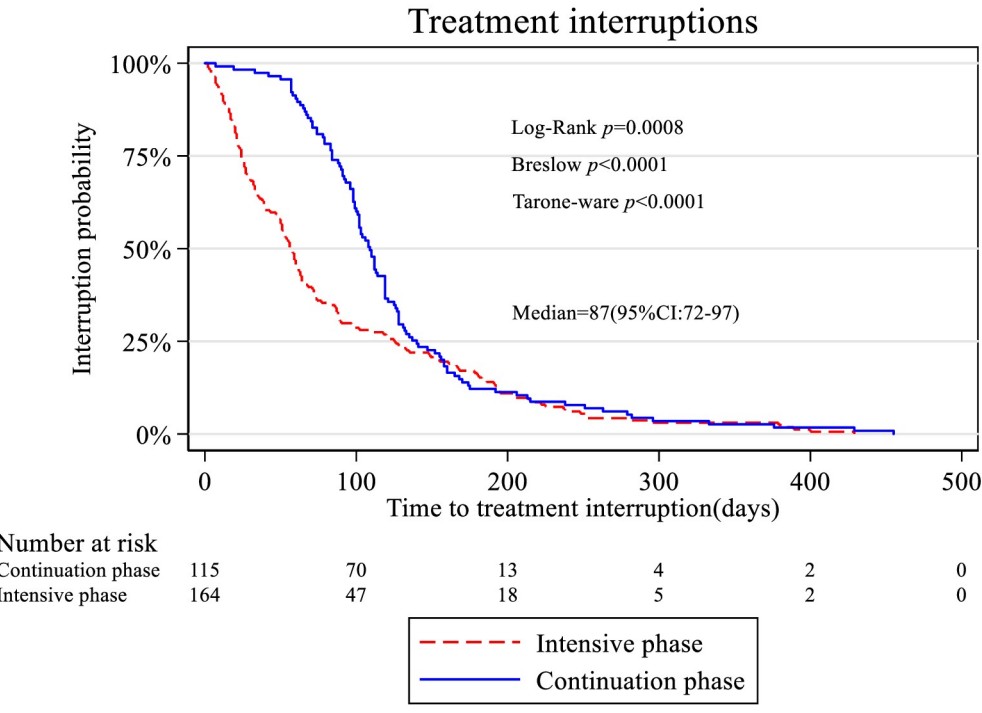

**Fig 3. Kaplan-Meir curves for the treatment phase.**

**Table 3. Cox regression estimates of the risk factors for the time to TB drug interruption in patients attending Kiambu County public hospitals (N = 292).**

| Characteristics | Unadjusted model* | | | | Adjusted model** | | | |
|---|---|---|---|---|---|---|---|---|
| | cHR | Std. Err | p-value | 95%CI | aHR | Std. Err | p-value | 95%CI |
| Sex | | | | | | | | |
| Female | ref | | | | ref | | | |
| Male | 0.997 | 0.128 | 0.981 | 0.775–1.283 | 1.052 | 0.156 | 0.714 | 0.802–1.381 |
| Age | 0.998 | 0.005 | 0.683 | 0.989–1.007 | 0.999 | 0.005 | 0.772 | 0.989–1.008 |
| Type of TB | | | | | | | | |
| PTB | ref | | | | ref | | | |
| EPTB | 0.822 | | 0.698 | 0.305–2.216 | 1.028 | 0.535 | 0.958 | 0.370–2.852 |
| BMI | | | | | | | | |
| Obese | | | | | ref | | | |
| Normal | 1.051 | 0.149 | 0.727 | 0.795–1.388 | 1.118 | 0.163 | 0.446 | 0.40–1.488 |
| Moderate | 1.331 | 0.356 | 0.284 | 0.788–2.248 | 1.436 | 0.394 | 0.187 | 0.839–2.457 |
| Overweight | 0.603 | 0.199 | 0.127 | 0.315–1.154 | 0.543 | 0.187 | 0.077 | 0.276–1.067 |
| Severe | 0.954 | 0.166 | 0.787 | 0.678–1.342 | 0.976 | 0.183 | 0.898 | 0.677–1.407 |
| HIV status | | | | | | | | |
| Negative | ref | | | | ref | | | |
| Positive | 1.03 | 0.136 | 0.822 | 0.795–1.335 | 1.133 | 0.159 | 0.374 | 0.860–1.493 |
| Phase | | | | | | | | |
| Continuation | ref | | | | ref | | | |
| Intensive | 1.504 | 0.186 | 0.001 | 1.181–1.916 | 1.581 | 0.202 | < 0.0001 | 1.232–2.031 |

interruption. This confers a great risk of continuous community TB transmission prior to conversion to smear negative [18].

Expectedly, majority of our study participants were HIV negative (70.9%). This reflects the TB prevalence survey that was conducted in Kenya in 2015–2016 which revealed that upto 85% of the population with TB in the country were HIV negative [5]. Despite being returned back to care and treatment, (33.1%) of the study participants had unfavorable outcomes with upto (20.9%) interrupting treatment yet again. This informs the need to come up with measures addressing the reasons cited for interruption in order to mitigate this pattern.

Our study revealed that 71.8% of the patients that interrupted treatment cited lack of knowledge for the missed clinic appointments. Inadequate information on the disease, duration of treatment, as well as risks associated with treatment interruption could contribute to non-adherence. This echoes previous studies that linked low level of knowledge to treatment interruption [19,20]. It is thus paramount to uphold patient health education as one of the key components in TB management. Increase in the level of knowledge on TB will consequently improve patients' health seeking behavior. In our study, patient relocation and travelling followed closely as reasons for interrupting treatment. Improving provider to patient relationship, as well as strengthening linkage mechanisms will ensure the patients have access treatment despite movement from the primary health facility.

Subsequent to TB treatment initiation, majority of the patients feel much better and get relief from the symptoms of TB. This is due to the bactericidal activity of the TB drugs which rapidly reduces the bacillary load in a patient [21]. However, this could also attribute to patients interrupting treatment. Previously studies conducted have associated feeling better as a contributor to treatment interruption [15,22,23]. Health education on importance of adherence, coupled with close patient follow-up could assist in ensuring patients are retained in care until treatment completion.

Treatment interruption was highest among males. Social behavior, highly mobile in nature, as well as poor health seeking behaviors could be a recipe that could contribute to non-adherence. Besides the prevalence survey in Kenya that showed males had a higher risk of acquiring TB, studies done have shown female gender to be more adherent to treatment unlike their counterparts [24]. Further, the age group most affected was 25–34 years. This is an age bracket that is economically productive hence likely to interrupt treatment as they work to sustain their livelihood.

Treatment interruption remains a major hurdle in the fight against TB. As the study has revealed, majority of the patients interrupt treatment during the intensive phase. It is apparent that some of the major reasons of treatment interruption are lack of knowledge, travel, relocation as well as feeling better. Therefore, health education and counseling on TB transmission, treatment duration, and importance to complete treatment and potential drug adverse effects could help in strengthening adherence to treatment. Improved health provider to patient relationship, having a reminder sms platform coupled with diarized patients appointments are ways that could help in reminding the patient of the clinic visits. Patient care centered approach will also help understand the preferred health facility the patient would want to seek care. This approach will also help in linkage for the patient in case of need to travel or relocation from the primary health facility.

## Strengths

The study setting is unique in that it encompasses both rural and urban settings thus allows for the analysis of the similarities and various differences of the factors influencing(hindering successful TB treatment among our patients) interruption depending on the patient locality. In

addition, our study incorporated LFTUs that were returned back to treatment within the TB treatment period (pre-TB treatment). This ensured that we were able to follow up the patients to ascertain their treatment outcomes. The period between the treatment interruption and tracing was within the pre-treatment period hence this reduced recall bias. Our study was able to incorporate participants from both public and private health facilities making it representative of the health facilities from whole county. The STROBE guidelines for cohort studies informed the design and reporting for this study. Finally, our study used a standardized MOH tool to capture details of treatment interruption that was filled by the CHV and verified by health worker to ensure completeness of documentation in the tracing form.

## Study limitation

Some of the study limitations found includes exclusion of some TB treatment interrupters that were not found during the defaulter tracing exercise as some facilities were not linked with CHVs, this had the potential to affect the number of patients included in the study as they were left out of the study. Some of the socioeconomic parameters that could affect patient treatment outcomes were not recorded (e.g. smoking status, alcohol consumption). The data collection method was retrospective and relied on the accuracy of data recorded, thus there is possibility that data were not recorded correctly were excluded in the study. Finally, majority of treatment interrupters in our study were men, this might have introduced a selection bias in the population included in our study.

## Conclusions

The study showed that treatment interruption is common among males, dominant in the population aged 25–34 years, and among those who stays in Kiambaa and Githunguri sub-counties. The treatment success rate was estimated to be 66.8% while lack of knowledge and relocation were the major reasons for the treatment interruptions. Patients on intensive phase of the treatment had higher risk of treatment interruption. To reduce the treatment interruption TB treatment program consider having interventions that target men to educate men on the dangers of treatment interruptions and middle aged group of the population in order to ensure treatment completion is embraced.

## Supporting information

**S1 File. Treatment interruption tracing form.**
(DOCX)

**S1 Data.**
(XLSX)

## Acknowledgments

We wish to acknowledge the support of Afya Bora Consortium for the scholarship they accorded me towards studying global health leadership. Also, we appreciate Dr Gachuno, Dr Theresa, Samuel Muhula, Titus Kiptai, and James Orwa for their guidance during the whole process of developing this manuscript. Finally, we are grateful for the Kiambu County, health department who allowed us to access the patients' files for data extraction.

## Author Contributions

**Conceptualization:** Evelyn Kimani, Onesmus Gachuno.

**Data curation:** Evelyn Kimani.

**Formal analysis:** Evelyn Kimani, Samuel Muhula, Titus Kiptai, James Orwa.

**Methodology:** Evelyn Kimani, Samuel Muhula, Titus Kiptai, James Orwa.

**Project administration:** Evelyn Kimani.

**Supervision:** Samuel Muhula, Titus Kiptai, Theresa Odero, Onesmus Gachuno.

**Validation:** Evelyn Kimani, Titus Kiptai.

**Visualization:** Evelyn Kimani, Samuel Muhula, James Orwa.

**Writing – original draft:** Evelyn Kimani.

**Writing – review & editing:** Evelyn Kimani, Samuel Muhula, Titus Kiptai, James Orwa, Theresa Odero, Onesmus Gachuno.

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
