## [Decision Letter · Decision Letter 0]

7 Jan 2021

PONE-D-20-33540

Factors influencing TB Treatment interruption and treatment outcomes among patients in Kiambu County, 2016-2019

PLOS ONE

Dear Dr. Evelyn Kimani,

Thank you for submitting your manuscript to PLOS ONE. After careful consideration, we feel that it has merit but does not fully meet PLOS ONE’s publication criteria as it currently stands. Therefore, we invite you to submit a revised version of the manuscript that addresses the points raised during the review process.

This is generally a well written paper but we ask that you respond to the editor and peer-reviewer's comments and revise your manuscript accordingly. Please ensure that a spell check is done as a number of grammatical errors have been noted. PLOS ONE does not copy edit accepted manuscripts. We therefore urge that a thorough spell check is done. Do also ensure that you go through the STROBE reporting checklist for observational studies to ensure that all relevant items are reported. Reporting according to STROBE guidelines could be added as a strength of your study in your manuscript. If done we ask that you attach this checklist when resubmitting your manuscript.

The manuscript strongly concludes that men are prone to treatment interruption compared to women yet the sample population included 68% men. Wouldn't including more men bias your study findings? We would appreciate this discussion in the limitation section of your manuscript.

In addition the peer reviewer has highlighted some changes that warrant your attention. These include a mismatch between statistical figures in the abstract and manuscript, encouragement to reference the latest WHO TB reports and Kenyan National Leprosy and TB program reports, misreporting of acronyms, consistency in the use of HR or aHR among other minor comments.

We look forward to receiving your revised manuscript.

Kind regards,

Eleanor Ochodo, M.D., PhD

Academic Editor

PLOS ONE

2.) You indicated that a waiver of consent was granted for your study,; however, you have not indicated whether the institutional review board waived the need for ethical approval. We understand that the framework for ethical oversight requirements for studies of this type may differ depending on the setting and we would appreciate some further clarification regarding your research. Could you please provide further details confirming from your institutional review board or research ethics committee (e.g., in the form of a letter or email correspondence) that ethics review was not necessary for this study? Please include a copy of the correspondence as an "Other" file."

3.) We suggest you thoroughly copyedit your manuscript for language usage, spelling, and grammar. If you do not know anyone who can help you do this, you may wish to consider employing a professional scientific editing service.  

4.) Please provide further details on sample size and power calculations.

5.) In the statistical analysis section, please clarify which confounders or variables were accounted for in your statistical models.

6.) Please include additional information regarding the data extraction tool used in the study and ensure that you have provided sufficient details that others could replicate the analyses. For instance, if you developed the data extraction tool as part of this study and it is not under a copyright more restrictive than CC-BY, please include a copy, in both the original language and English, as Supporting Information, or include a citation if it has been published previously.

7.) We note that you have indicated that data from this study are available upon request. PLOS only allows data to be available upon request if there are legal or ethical restrictions on sharing data publicly. For information on unacceptable data access restrictions, please see http://journals.plos.org/plosone/s/data-availability#loc-unacceptable-data-access-restrictions.

8.) Your ethics statement should only appear in the Methods section of your manuscript. If your ethics statement is written in any section besides the Methods, please move it to the Methods section and delete it from any other section. Please ensure that your ethics statement is included in your manuscript, as the ethics statement entered into the online submission form will not be published alongside your manuscript.

9.) Please include your tables as part of your main manuscript and remove the individual files. Please note that supplementary tables (should remain/ be uploaded) as separate "supporting information" files

Reviewers' comments:

Reviewer's Responses to Questions

**Comments to the Author**

1. Is the manuscript technically sound, and do the data support the conclusions?

Reviewer #1: Yes

2. Has the statistical analysis been performed appropriately and rigorously? 

Reviewer #1: Yes

3. Have the authors made all data underlying the findings in their manuscript fully available?

Reviewer #1: Yes

4. Is the manuscript presented in an intelligible fashion and written in standard English?

Reviewer #1: Yes

5. Review Comments to the Author

Reviewer #1: I have appended minor comments onto the manuscript for your action. In summary, the numbers in the results section and abstracts should match. I realize you used HR instead of aHR in the abstract section. Addition of latest reports such as WHO TB Report 2020 and Annual TB Program Report 2020 would greatly suffice in the manuscript. Use of acronyms without declaration is not best writing etiquette. I asked whether CHVs and CHEWs are synonymous or exclusive in the field work that was conducted as part of the study.

6. PLOS authors have the option to publish the peer review history of their article (what does this mean?). If published, this will include your full peer review and any attached files.

Reviewer #1: **Yes: **Dr. Steve Wandiga

---

## [Author Response · Author response to Decision Letter 0]

15 Feb 2021

Dear Editor,

Thank you for your review. We have addressed the comments as highlighted in blue. Thank you. 

 Thank you. Yes, this was checked; the manuscript conforms to the PLOS One authors submission guidelines

2.) You indicated that a waiver of consent was granted for your study,; however, you have not indicated whether the institutional review board waived the need for ethical approval. We understand that the framework for ethical oversight requirements for studies of this type may differ depending on the setting and we would appreciate some further clarification regarding your research. Could you please provide further details confirming from your institutional review board or research ethics committee (e.g., in the form of a letter or email correspondence) that ethics review was not necessary for this study? Please include a copy of the correspondence as an "Other" file."

 Yes there was a waiver; we have included a section on ethical consideration on page 7 and attached the ethical approval letter as an additional file (S2). 

3.) We suggest you thoroughly copyedit your manuscript for language usage, spelling, and grammar. If you do not know anyone who can help you do this, you may wish to consider employing a professional scientific editing service. 

 Thank you this has been checked, the grammar and the spelling has been checked.

4.) Please provide further details on sample size and power calculations.

 There was no formal sample size calculation as this was a retrospective study, however the final sample included in the study was assumed to have enough power to detect a difference between the outcome and explanatory variables. This is explained in page 6.

5.) In the statistical analysis section, please clarify which confounders or variables were accounted for in your statistical models.

There is a section added on study variables on page 7. All variables were considered important in explaining the outcome of interest and were therefore adjusted for in the cox proportion model.

6.) Please include additional information regarding the data extraction tool used in the study and ensure that you have provided sufficient details that others could replicate the analyses. For instance, if you developed the data extraction tool as part of this study and it is not under a copyright more restrictive than CC-BY, please include a copy, in both the original language and English, as Supporting Information, or include a citation if it has been published previously.

We used a treatment interrupters tracer form which is a Ministry of Health approved tool to capture the participant’s details and reasons for treatment interruption; this has been indicated in page 5. This tool has been attached as an additional file (S1)

7.) We note that you have indicated that data from this study are available upon request. PLOS only allows data to be available upon request if there are legal or ethical restrictions on sharing data publicly. For information on unacceptable data access restrictions, please see http://journals.plos.org/plosone/s/data-availability#loc-unacceptable-data-access-restrictions.

 The data set has been uploaded as an additional file, S3.

8.) Your ethics statement should only appear in the Methods section of your manuscript. If your ethics statement is written in any section besides the Methods, please move it to the Methods section and delete it from any other section. Please ensure that your ethics statement is included in your manuscript, as the ethics statement entered into the online submission form will not be published alongside your manuscript.

Thank you. The Ethics section has been added to the methods section page 7.

9.) Please include your tables as part of your main manuscript and remove the individual files. Please note that supplementary tables (should remain/ be uploaded) as separate "supporting information" files

 The tables have now been included as part of the main manuscript.

 The STROBE guidelines for cohort studies informed the design and reporting for this study. This has been added as additional file, S3.

The manuscript strongly concludes that men are prone to treatment interruption compared to women yet the sample population included 68% men. Wouldn't including more men bias your study findings? We would appreciate this discussion in the limitation section of your manuscript.

Majority of treatment interrupters in our study were men, this might have introduced a selection bias in the population included in our study. This statement has been added in page 14.

In addition the peer reviewer has highlighted some changes that warrant your attention. These include a mismatch between statistical figures in the abstract and manuscript, encouragement to reference the latest WHO TB reports and Kenyan National Leprosy and TB program reports, misreporting of acronyms, consistency in the use of HR or aHR among other minor comments.

Thank you, this has been corrected. We used the current reports as at the time of developing this study. 

Yours sincerely,

Evelyn Kimani

---

## [Decision Letter · Decision Letter 1]

8 Mar 2021

Factors influencing TB Treatment interruption and treatment outcomes among patients in Kiambu County, 2016-2019

PONE-D-20-33540R1

Dear Dr. Evelyn Kimani,

We’re pleased to inform you that your manuscript has been judged scientifically suitable for publication and will be formally accepted for publication once it meets all outstanding technical requirements.

Kind regards,

Eleanor Ochodo, M.D., PhD

Academic Editor

PLOS ONE

Additional Editor Comments (optional):

Reviewers' comments:

Reviewer's Responses to Questions

**Comments to the Author**

1. If the authors have adequately addressed your comments raised in a previous round of review and you feel that this manuscript is now acceptable for publication, you may indicate that here to bypass the “Comments to the Author” section, enter your conflict of interest statement in the “Confidential to Editor” section, and submit your "Accept" recommendation.

Reviewer #1: All comments have been addressed

2. Is the manuscript technically sound, and do the data support the conclusions?

Reviewer #1: Yes

3. Has the statistical analysis been performed appropriately and rigorously? 

Reviewer #1: Yes

4. Have the authors made all data underlying the findings in their manuscript fully available?

Reviewer #1: Yes

5. Is the manuscript presented in an intelligible fashion and written in standard English?

Reviewer #1: Yes

6. Review Comments to the Author

Reviewer #1: The paper meets the review comments and there are no additional comments for the author and there are no concerns about dual publication, research ethics, or publication ethics from my end.

7. PLOS authors have the option to publish the peer review history of their article (what does this mean?). If published, this will include your full peer review and any attached files.

Reviewer #1: **Yes: **Dr. Steve Wandiga

---

## [Editor Report · Acceptance letter]

22 Mar 2021

PONE-D-20-33540R1 

Factors influencing TB Treatment interruption and treatment outcomes among patients in Kiambu County, 2016-2019 

Dear Dr. Kimani:

I'm pleased to inform you that your manuscript has been deemed suitable for publication in PLOS ONE. Congratulations! Your manuscript is now with our production department. 

Kind regards, 

on behalf of

Dr. Eleanor Ochodo 

Academic Editor

PLOS ONE